# Differential miRNA and Protein Expression Reveals miR-1285, Its Targets TGM2 and CDH-1, as Well as CD166 and S100A13 as Potential New Biomarkers in Patients with Diabetes Mellitus and Pancreatic Adenocarcinoma

**DOI:** 10.3390/cancers16152726

**Published:** 2024-07-31

**Authors:** Theodoros Kolokotronis, Britta Majchrzak-Stiller, Marie Buchholz, Vanessa Mense, Johanna Strotmann, Ilka Peters, Lea Skrzypczyk, Sven-Thorsten Liffers, Louise Massia Menkene, Mathias Wagner, Matthias Glanemann, Fay Betsou, Wim Ammerlaan, Ronny Schmidt, Christoph Schröder, Waldemar Uhl, Chris Braumann, Philipp Höhn

**Affiliations:** 1St. Josef Hospital Bochum, Surgical Clinic, Ruhr-University Bochum, Gudrunstr. 56, 44791 Bochum, Germany; britta.majchrzak-stiller@ruhr-uni-bochum.de (B.M.-S.); marie.buchholz-a7y@rub.de (M.B.); vanessa.mense@studmail.w-hs.de (V.M.); johanna.strotmann@ruhr-uni-bochum.de (J.S.); ilka.peters@rub.de (I.P.); lea.skrzypczyk@klinikum-bochum.de (L.S.); waldemar.uhl@ruhr-uni-bochum.de (W.U.); chris.braumann@evk-ge.de (C.B.); philipp.hohn@ruhr-uni-bochum.de (P.H.); 2Institute of Pathology and Surgical Clinic, University Hospital of Saarland, Kirrberger Str. 100, 66424 Homburg, Germany; louise.massia@hotmail.fr (L.M.M.); mathias.wagner@uks.eu (M.W.);; 3University Hospital Essen, Bridging Institute for Experimental Tumor Therapy, West German Tumor Center Essen, Hufelandstr. 55, 45147 Essen, Germany; sven-thorsten.liffers@uk-essen.de; 4CRBIP, Institut Pasteur, Université Paris Cite, 25 rue du Dr Roux, 75015 Paris, France; fay.betsou@pasteur.fr; 5IBBL (Integrated BioBank of Luxembourg), 1, Rue Louis Rech, L-3555 Dudelange, Luxembourg; wim.ammerlaan@ibbl.lu; 6Sciomics GmbH, Karl-Landsteiner Str. 6, 69151 Heidelberg, Germany; ronny.schmidt@sciomics.de (R.S.); christoph.schroeder@sciomics.de (C.S.); 7Department of General, Visceral and Vascular Surgery, EvK Gelsenkirchen, University Duisburg-Essen, Munckelstr. 27, 45879 Gelsenkirchen, Germany

**Keywords:** pancreatic adenocarcinoma, biomarkers, diabetes mellitus, S100A13, miR-1285

## Abstract

**Simple Summary:**

Pancreatic ductal adenocarcinoma is a very lethal tumor entity with the late appearance of clinical signs and the absence of biomarkers for early diagnosis. Diabetes mellitus can increase the risk of pancreatic cancer, as well as the effect of treatment modalities, so patients with PDAC and DM represent a special study group. We validated miRNA1285-3p, TGM2, CDH-1, CD166, and S100A13 as potential meaningful biomarkers to characterize patients with PDAC + DM. The present study represents a pilot approach to address this question. The potential biomarkers presented in the current study could contribute to the search for biomarkers for the diagnosis of PDAC, enabling early detection of PDAC in DM patients in the future.

**Abstract:**

Early detection of PDAC remains challenging due to the lack of early symptoms and the absence of reliable biomarkers. The aim of the present project was to identify miRNA and proteomics signatures discriminating PDAC patients with DM from nondiabetic PDAC patients. Proteomics analysis and miRNA array were used for protein and miRNA screening. We used Western blotting and Real-Time Quantitative Reverse Transcription polymerase chain reaction (qRT-PCR) for protein and miRNA validation. Comparisons between experimental groups with normal distributions were performed using one-way ANOVA followed by Tukey’s post hoc test, and pairwise tests were performed using *t*-tests. *p* ≤ 0.05 was considered statistically significant. Protein clusters of differentiation 166 (CD166), glycoprotein CD63 (CD63), S100 calcium-binding protein A13 (S100A13), and tumor necrosis factor-β (TNF-β) were detected in the proteomics screening. The miRNA assay revealed a differential miRNA 1285 regulation. Previously described target proteins of miR-1285 cadherin-1 (CDH-1), cellular Jun (c-Jun), p53, mothers against decapentaplegic homolog 4 (Smad4), human transglutaminase 2 (TGM2) and yes-associated protein (YAP), were validated via Western blotting. miR-1285-3p was successfully validated as differentially regulated in PDAC + DM via qRT-PCR. Overall, our data suggest miRNA1285-3p, TGM2, CDH-1, CD166, and S100A13 as potential meaningful biomarker candidates to characterize patients with PDAC + DM. Data are available via ProteomeXchange with the identifier PXD053169.

## 1. Introduction

PDAC is a frequent tumor entity with remarkably high lethality due to late diagnosis, tumor aggressiveness, and early appearance of distal metastases. The five-year survival of patients with PDAC is 10%; therefore, PDAC is the tumor entity with the second worst survival rate in Germany [1]. Similar five-year survival rates have been reported in the United States of America [2]. It has been suggested that PDAC will become the second leading cause of cancer-related death by 2030 [1].

While surgical resection offers a clear survival benefit and increases the 5-year survival to 25%, most patients suffer unfortunately from disseminated disease at the time of first diagnosis, so systemic chemotherapy is the only therapeutic option. The most frequently used therapeutic agent is Gemcitabine, a drug that the FDA first approved for metastatic PDAC in 1996 [3]. The multi-drug regimen FOLFIRINOX (5-Fluorouracil, Leucovorin, Irinotecan, and Oxaliplatin) has also shown efficacy in metastatic PDAC. FOLFIRINOX offers improved disease-free survival compared to Gemcitabine (21.6 vs. 12.8 months), though FOLFIRINOX is associated with a higher rate of serious adverse effects (75.9 vs. 52.9%) [4]. Recently, it has been suggested that neoadjuvant therapy with FOLFIRINOX improves overall survival (OS) in patients with borderline resectable PDAC [5]. Randomized controlled trials are currently being conducted to address this question [6].

Aside from standard chemotherapy treatments, different targeted therapies based on immune checkpoint inhibitors (ICIs) and small-molecule kinase inhibitors (SMKIs) have been reported. Hyperactivated focal adhesion kinase activity (FAK) was elevated in human PDAC tissues and correlated with elevated levels of fibrosis and poor CD8+ cytotoxic T-cell infiltration. Single-agent FAK inhibition using the selective FAK inhibitor VS-4718 substantially limited tumor progression, resulting in a doubling of survival in the p48-Cre;LSL-KrasG12D;Trp53flox/+ (KPC) mouse model of human PDAC [7]. 1,3,4-Oxadiazole and 1,3,4-Thiadiazole Nortopsentin derivatives demonstrated a CDK1 Inhibition in PDAC cell lines. CDK1 is a crucial regulator of cell-cycle progression and cancer cell proliferation [8]. Moreover, 3-Amino-1,2,4-Triazine PDK inhibitors exhibit cytotoxic effects on 2D and 3D PDAC cell models, thus encouraging further structure manipulation for the development of analogs against PDAC [9].

In 90% of all PDAC cases, the proto-oncogenic Kirsten rat sarcoma viral oncogene homolog (short: KRAS) has mutated [10]. The KRAS gene encodes a Guanosine Triphosphate (GTP)ase transductor protein, involved in the regulation of cell division by transmitting external signals to the cell nucleus [10,11]. Mutations in the KRAS gene lead to the activation of the p53 and p16 pathways, resulting in turn in senescence or cell-cycle arrest if no further mutations in these oncogenes are present. This mechanism is a natural protection against cancer [10,11,12]. In 50–70% of cases of PDAC, the TP53 gene is deactivated by a mutation and can, therefore, no longer express p53. In 95% of cases, the CDKN2A gene is deactivated by a mutation, which leads to a lack of p16 expression. A commutation of p53 and p16, accompanied by an activating mutation of KRAS, results in uncontrolled activation of further downstream pathways, such as the AKT pathway. This, in turn, causes increased proliferation and cell survival, e.g., via the mechanistic target of the rapamycin (short: mTOR) pathway [10,13,14].

There is a lack of biomarkers for the early diagnosis of PDAC. One possibility to detect cancer is liquid biopsy. The most promising biomarker types in liquid biopsy are cell-free DNA, circulating tumor cells, exosomes, and microRNAs (short: miRNAs) [15]. Due to the low time, cost consumption, and invasiveness, liquid biopsy could be used as a screening tool in selected patients [15,16]. In particular, the measurement of tumor-derived miRNAs in serum or plasma is an important approach for the blood-based detection of human cancer [17], where miRNA profiles differ significantly between organs, and circulating miRNomics in plasma represent these organ-specific cancers [18].

The general risk of developing cancer is increased in patients with a history of DM. Concerning type 1 diabetes mellitus (T1DM), studies link the disease to an elevated risk of leukemia, pancreatic, liver, mouth and pharyngeal, stomach, skin, and ovarian cancer, although these studies are often not consistent. Additionally, the distinction between T1DM and type 2 diabetes mellitus (T2DM) tends to be vague [19]. Patients with T2DM have an increased risk of developing cancers such as liver, pancreatic, endometrial, colorectal, breast, and bladder cancer [20]. In the case of pancreatic cancer, 50–80% of all patients present with diabetes mellitus (DM), and 85% of these diabetes diagnoses were made less than 2 years prior to the pancreatic cancer diagnosis [21]. This suggests a close link between these two diseases. On the one hand, long-term DM increases the pancreatic cancer risk by 1.5 to 2-fold, and PDAC patients with DM tend to have larger tumors and worse prognosis [21]. However, a recent retrospective study including 2643 pancreatic ductal adenocarcinoma patients undergoing pancreatic head resection from 2013 to 2017 within the German pancreatic surgery registry (DGAV StuDoQ|Pancreas) showed that new-onset diabetes was associated with better survival [22]. On the other hand, pancreatic cancer impairs pancreatic function, therefore causing pancreatogenic DM, so-called type 3c DM. Thus, T3cDM can be an early sign of pancreatic cancer [20,23]. The underlying mechanism by which DM increases pancreatic cancer risk is complex, but it is known that metabolic, hormonal, and immunological alterations play important roles [20]. New-onset diabetes, as an early symptom of PDAC, can frequently be misdiagnosed [24]. Unfortunately, there is a lack of diagnostic markers to detect DM as an early PDAC symptom, except for CA 19-9 [25].

The aim of the present project was to investigate whether patients with PDAC + DM have differentiated miRNA and protein expression that could be used as screening for early PDAC diagnosis in this group of patients.

## 2. Materials and Methods

### 2.1. Patient Recruitment and Sample Collection

After approval from the local Ethics Review Board (Number 105/16, ethics committee of the Medical Association of Saarland, Germany), blood plasma and tissue samples from patients with pancreatic disease who presented for surgical therapy at the University Clinic of Saarland (UKS) were collected, between 1 September 2016 and 31 December 2017. Written informed consent was mandatory for all patients. Nineteen patients with PDAC were recruited for the study. Tissue samples were collected intraoperatively from the pancreatic tumor area. Patients with other types of disease in the final histological finding (chronic pancreatitis, distal cholangiocellular adenocarcinoma, etc.) were excluded from further analysis. Diabetic patients without PDAC were also excluded from further analysis because only blood samples could be obtained from this subgroup of patients, as no surgery was needed. Clinical, pathological, and survival data were documented. The analyses were performed retrospectively. Blood samples were collected and processed for plasma isolation and RNA lysate extraction. Blood samples were taken preoperatively (1 to 5 days before pancreatic surgery). Samples containing 500 μL of plasma were prepared from this blood and deep-frozen. HbA1C and CA19-9 levels were determined at a centralized certified clinical laboratory and documented.

### 2.2. Proteomics Screening Analysis

As screening to evaluate for differentiated protein expression, six tissue samples from PDAC patients with and without DM were submitted to proteomics analysis at the Sciomics Research Institute in Heidelberg, Germany. The samples were analyzed in a dual-color approach using a reference-based design on 6 Scio Discover antibody microarrays (Sciomics, Heidelberg, Germany) targeting 1360 different proteins with 1830 antibodies. Each antibody was represented on the array in four replicates. The arrays were blocked with Scio Block (Sciomics) on a Hybstation 4800 (Tecan, Grödig, Austria), and afterward, the samples were incubated competitively using a dual-color approach. After incubation for three hours, the slides were thoroughly washed with 1× PBSTT, rinsed with 0.1× PBS as well as with water, and subsequently dried with nitrogen. Slide scanning was conducted using a power scanner (Tecan, Austria) with identical instrument laser power and adjusted PMT settings. Spot segmentation was performed with GenePix Pro 6.0 (Molecular Devices, Union City, CA, USA). Acquired raw data were analyzed using the linear models for the microarray data (LIMMA v.3.60.4) package of R-Bioconductor after uploading the median signal intensities. For normalization, a specialized invariant Lowes’s method was applied. For analysis of the samples, a one-factorial linear model was fitted with LIMMA, resulting in a two-sided *t*-test or F-test based on moderated statistics. All presented *p*-values were adjusted for multiple testing by controlling the false discovery rate according to Benjamini and Hochberg. Proteins were defined as a differential for logFC value > 0.5 and an adjusted *p*-value < 0.05. Differences in protein abundance between different samples or sample groups are presented as log-fold changes (logFC) calculated for basis two (for example, logFC = 1: sample group = 2-fold higher signal as the control group; logFC = −1 sample group 1/2 of the signal compared to the control group).

### 2.3. Differentiated miRNA Expression Analysis in Blood Plasma Samples

#### 2.3.1. miRNA Extraction from Plasma

For miRNA extraction from the plasma of 19 samples, the Qiagen miRNA easy advanced serum plasma kit and the semiautomatic QIAcube system (Qiagen) were used following the manufacturer’s instructions. For a single extraction with the QIAcube, a maximum of 200 µL of plasma was used to keep the level of impurities as low as possible. Out of a 500 µL volume of plasma provided by each patient, 2 extractions of 200 µL each were used. Both were eluted in nuclease-free water and poled afterward. One validated sample, “IPC_Plasma”, and the “miRNeasy Serum/Plasma Spike-In Control” served as controls.

#### 2.3.2. miRNA Quantification

To quantify a miRNA, the Synergy Mx Monochromator-based multimode microplate reader and the corresponding BioTek Gen5 v.2.09 data analysis software were used for two different methods. To determine the RNA concentration and purity of the 44 RNA solutions, they were initially measured by spectrophotometry at 230 nm, 260 nm, and 280 nm, and the quotients 260/280 and 260/230 were calculated. Additionally, the RNA concentration was determined by spectrofluorometry using the ultrasensitive fluorescent indicator RiboGreen (RiboGreen™ RNA quantification kit, Invitrogen, Darmstadt, Germany). After quantification, the RNA extracts were concentrated by water evaporation and finally dissolved in 5 μL of nuclease-free water. Calculations were performed automatically by the BioTek Gen5 v.2.09 data analysis software.

#### 2.3.3. miRNA Profiling Using a Takara Platform

For miRNA profiling, the Wafergen Smart Chip Human miRNA panel v3.0 from Takara was used. A total of 19 blood plasma samples from PDAC patients with and without DM were analyzed for differentiated miRNA expression. This step was performed in collaboration with the Integrated Biobank of Luxembourg (IBL). In this miRNA microarray, more than 1000 potential miRNA candidates were compared between PDAC patients with and without DM. First, the extracted miRNA from the plasma samples was diluted with nuclease-free water. The RNA concentration of the resulting solutions was measured via spectrophotometry and spectrofluorometry. Afterward, the miRNA was polyadenylated to be converted to cDNA via reverse transcription. Finally, the nucleic acid was amplified on chips by real-time quantitative polymerase chain reaction (qPCR), allowing the study of miRNA expression profiles. Differential miRNA expression was calculated with the bioconductor package Limma (v.3.44.3). Therefore, miRNAs not expressed in at least one group were excluded from further analysis, and missing values were substituted with a Ct-value of 33, followed by the estimation of detected molecules/µL, which were log2-transformed and quantile normalized. A linear model was fitted to the normalized data, and differential expression was assessed by computing the empirical Bayes statistics.

### 2.4. miRNA Validation, cDNA Synthesis from Plasma Samples, and Quantitative Real-Time PCR

The selection of the patients from the collective was based on the histology of the tissue sample and the presence of DM. A total of 19 patients were selected for whom tissue samples and serum were available and who were classified within one of the two disease groups—PDAC or PDAC + DM (Table 1). RNA lysates obtained from blood plasma from the selected patients were stored in the biodatabase.

For cDNA synthesis via reverse transcription (RT), the TaqMan Advanced miRNA cDNA synthesis kit from Thermo Fisher Scientific (Dreieich, Germany) was used following the manufacturer’s instructions, where 5 ng/μL RNA from the previously isolated RNA lysates was applied. The reverse transcription of the miRNAs takes place with universal primers that bind to the poly(A) tail (poyATailing 37 °C, 15 min; rev. transcription 42 °C, 15 min).

For the real-time quantitative polymerase chain reaction (qPCR), the TaqMan Advanced miRNA Assay from Thermo Fisher Scientific was used according to the manufacturer’s protocol. In this qPCR, the desired cDNA products were amplified and quantified according to the TaqMan principle using hsa-miR-1285-5p Assay ID: 479565_mir, hsa-miR-1285-3p Assay ID: 478687_mir, hsa-miR-16-5p Assay ID: 477860_mir and hsa-miR-21-5p Assay ID: 477975_mir (95 °C, 5 min Enzyme activation; 95 °C, 3 s Denature and 60 °C, 30 s Annealing/Extending for 14 cycles; 99 °C, 10 min Sopp). miRNA regulation was then determined using the ΔΔCt method. Normalization was performed using the housekeeping miRNA miR-16-5p.

### 2.5. Protein Targets of miRNA-1285 in the Literature

Apart from the above potential candidate protein markers characterizing the PDAC + DM group, literature research was performed concerning already described protein targets of miRNA-1285. The following protein candidate targets were identified: CDH-1 [26], c-Jun [27], p53 [28], Smad4 [26], TGM2 [29] and YAP [30,31].

### 2.6. Tissue Lysis and Protein Expression Analysis with Western Blot for Protein Target Validation

Validation of the differential protein markers was performed on the previously analyzed patient collective. Patients were selected from the collective (Table 1) based on tissue sample histology and the presence of DM. A total of 13 patients were selected who were classified within one of the two disease groups PDAC and PDAC + DM. Tissue lysis with RIPA buffer and BCA test: The radioimmunoprecipitation assay (RIPA) buffer was used as one lysis buffer for cell disruption and solubilization of membranes supplemented with protease inhibitor (Protease Inhibitor Cocktail, Promega, 1×, Walldorf, Germany) and phosphatase inhibitor (PhosSTOP™, Roche, 1×, Mannheim, Germany) to prevent protein degradation and phosphatase activity. For homogenization, 30 mg of tissue RIPA in the Tissue Lyser LT from Qiagen using the according metal balls was used for 25–30 min at an oscillation frequency of 50 L/s. Then, the samples were incubated on ice for 10 min so that as many proteins as possible were released from the tissue debris. The cell debris was separated from the lysate in the supernatant by centrifuging the samples for 15 min at 13,200 rpm and 4 °C. The lysate was aliquoted and stored at −80 °C.

The protein concentration was determined using the bicinchoninic acid (BCA) test in duplicate according to the manufacturer’s protocol for the Pierce™ BCA Protein Assay Kit. To determine the concentration, a standard series with the bovine serum albumin (BSA) supplied by the kit, was used according to the manufacturer’s protocol.

Lysing of the tissue with titin sample buffer and determination of the protein concentration using a 660 nm protein assay: The second protocol used was originally designed for the lysis of the muscle protein titin from heart tissue [26]. The titin sample buffer contains SDS for dissolving the protein complexes and dithiothreitol (DTT) unfolding the proteins and achieving a uniform negative charge. The titin sample buffer is also a stained loading buffer, eliminating the need for sample preparation prior to SDS-PAGE. A small piece of tissue was manually homogenized within the titin sample buffer by cutting followed by 20 min of incubation on ice. After incubation, the samples were heated at 96 °C for 3 min, and the cell debris was spun down at 10,000 rpm for 10 min.

As the lysates were generated with the stained titin sample buffer, the PierceTM 660 nm Protein Assay Reagent was used in combination with the Ionic Detergent Compatibility Reagent (IDCR). For the 660 nm protein assay, a sufficient volume of the PierceTM 660 nm Protein Assay Reagent was mixed with the appropriate amount of IDCR (1 g per 20 mL reagent). On a microtiter plate, the samples and a BSA standard series were spiked with 200 μL of assay reagent. The microtiter plate was incubated for 10 min, and the absorbance was measured at a wavelength of 660 nm using an Infinite^®^ 200 PRO microplate reader from Tecan. The protein concentration was calculated using the BSA standard series.

SDS polyacrylamide gel electrophoresis: With sodium dodecyl sulfate (SDS), polyacrylamide gel electrophoresis (PAGE) proteins were separated according to their molecular weight under denaturing conditions. A total of 30 mg of protein from each sample was mixed with 5 μL of loading buffer (Roti-Load 1, Carl Roth, Karlsruhe, Germany) and incubated at 95 °C for 5 min. No sample preparation was necessary for the lysates prepared according to the protocol with titin sample buffer. Due to the lower protein concentration of these lysates, only 18 μg of these samples were applied to each well. Stain-free Mini-TGX gels (Bio-Rad Laboratories, Feldkirchen, Germany) were used for separation. The separation was conducted at 250 V with 1× running buffer.

Western blot with normalization over total protein: Proteins were detected and quantified using Western blot. To make a quantitative statement about protein expression, the samples were normalized to the total proteins present in each sample, using the Bio-Rad stain-free system according to the manufacturer’s instructions. After transferring the proteins to a PVDF membrane (0.2 μm) using the Trans-Blot^®^ TurboTM Transfer System (Bio-Rad), free binding sites were blocked with EveryBlot Blocking Buffer (Bio-Rad Laboratories, Feldkirchen, Germany) for 5 min at room temperature on an incubation shaker. The primary antibody was added to the blocking buffer (c-Jun mAb #9165 [60A8], E-cadherin mAb #3195T [24E10], p53 mAb #2527 [7F5], Smad4 mAb #38454 [D3M6U], TGM2 XP mAb #3557T [D11A6], and YAP XP^®^ mAb #14074 [D8H1X] from CST Europe, Frankfurt am Main, Germany and CD63 mAb ab271286 [KILL150A], CD166 mAb ab233750 [3D9F1], S100A13 mAb ab109252 [EPR4510], and TNF-β pAb (LTA) ab227929 from Abcam, Cambridge, UK). The membrane was incubated with the primary antibody at 4 °C overnight and washed with Tris-buffered saline containing Tween 20 (TBST). Incubation of the membrane with secondary antibody (anti-rabbit IgG, HRP-linked #7074 or anti-mouse IgG, HRP-linked #7076 from CST Europe, Frankfurt am Main, Germany) which was previously diluted in blocking buffer, was conducted at room temperature for one hour. The membrane was washed with TBST. After covering the membrane with enhanced chemiluminescence (ECL) or ECL Max substrate (Clarity or Clarity Max ECL Western Blotting Substrates, Bio-Rad Laboratories, Feldkirchen, Germany), the chemiluminescence reaction catalyzed by HRP was detected on a ChemiDocTM imaging system (Bio-Rad Laboratories, Feldkirchen, Germany). During normalization, the frequency of the protein to be examined was compared to the total protein detected. Normalization was performed using Bio-Rad Image Lab 6.1 software.

### 2.7. Statistics and Calculations

The results are expressed as the means ± SD. Comparisons between experimental groups with normal distributions were performed using one-way ANOVA followed by Tukey’s post hoc test, and pairwise tests were performed using unpaired *t*-tests. *p* ≤ 0.05 was considered statistically significant and is indicated in the figures as follows: * *p* ≤ 0.05; ** *p* ≤ 0.01; *** *p* ≤ 0.001.

## 3. Results

### 3.1. Patient Recruitment and Sample Collection

From 1 September 2016 to 31 December 2017, 19 patients were admitted for operative treatment because of PDAC (Table 1). This group includes all patients subjected to pancreatic resection who were diagnosed with ductal adenocarcinoma of the pancreas (PDAC) based on postoperative histological findings. A total of 6 out of 19 patients (31.6%) with PDAC suffered from DM at the time of hospital admission, of whom four patients suffered from insulin-dependent DM (Table 1).

Eight patients were intraoperatively diagnosed with unresectable PDAC, where only tumor biopsies were performed. Five out of eight patients (62.5%) with unresectable PDAC suffered from DM at the time of hospital admission, of whom three patients suffered from insulin-dependent DM (Table 1). Significantly higher CA 19-9 values were observed in this subgroup of patients compared with the subgroup of patients with resectable PDAC.

Seventeen patients subjected to curative pancreatic resection were diagnosed with other entities based on postoperative histological findings: ten patients with chronic pancreatitis (CP), six patients with distal cholangiocellular carcinoma (CCC), and one patient with neuroendocrine pancreatic tumor (Table 1). Seven out of ten patients (70%) with CP suffered from DM at the time of hospital admission, of whom three patients suffered from insulin-dependent DM. Measured CA 19-9 values were, respectively, low in this subgroup of patients (Table 1).

Gender, BMI, and HbA1C values are also registered in Table 1.

### 3.2. Proteomics Screening Analysis and Selection of Protein Targets for Validation

From group 1 samples (PDAC + DM) and group 2 samples (PDAC), 171 antibodies recorded a differential protein abundance. The results of the statistical analysis are listed in Appendix A.

The proteomics screening and the following STRING analysis and pathway classification (Figure 1, Table 2) revealed four proteins identified as appropriate potential protein candidates for further validation. The proteins CD166 and CD63 were highly regulated in the PDAC + DM group, whereas the proteins S100A13 and TNF-β were downregulated (Table 3).

### 3.3. miRNA Microarray Results and Selection of miRNA Targets

After statistical analysis, four miRNAs, miRNA-1285, miRNA-502-3p, miRNA-493*, and miRNA-185, were significantly regulated in patients with resected PDAC vs. PDAC + DM (Table 4, Figure 2).

Interestingly, miRNA-1285 was also significantly regulated in patients with no resectable PDAC, contrary to patients with no resectable PDAC + DM, as shown in Table 5.

miRNA-1285 with a logFC of 1.25 was selected as the most promising miRNA for validation, as it was significantly expressed in both subgroups of patients with resected and unresectable PDAC. Its direct target proteins c-Jun, p53, TGM2, YAP, CDH-1, and Smad4 were identified via literature research as potential protein markers for further validation [10,32] (Table 6).

### 3.4. Results of miRNA Validation via qRT-PCR in Blood Plasma Samples of the Patient Collective

The expression of miRNA-1285 was quantified using the advanced miRNA Assay System via qRT-PCR to visualize the differential regulation of miR-1285-3p and miR-1285-5p in patients with PDAC and PDAC + DM. Normalization was performed using the housekeeping miRNA miR-16-5p. Figure 3 shows the relative change in expression of the miRNA variants miR-1285-5p and miR-1285-3p. The changes in the individual ΔCt values of the patient samples are also shown in Figure 3. It is noticeable that the expression of miR-1285-5p is clearly higher than the expression of the 3p variant in all patients, regardless of their disease. The expression of miR-1285-3p varied between the diverse groups. Figure 3 shows the expression of each variant in the diverse groups of patients. miR-1285-5p was slightly, but not significantly reduced in the PDAC + DM disease groups compared to the PDAC group. For miR-1285-3p, a significantly lower expression was detected in PDAC + DM compared to PDAC (fold change 0.18) (Figure 3).

### 3.5. Results of Protein Validation/Target Protein Analysis in Tissue Samples via Western Blot

The differential protein markers that were identified as promising protein targets by the proteomics analysis: CD63, CD166, S100A13, and TNF-β, as well as the target proteins of the differentially expressed miRNA-1285: c-Jun, p53, CDH-1, Smad4, TGM2, and YAP, were analyzed/validated via Western blotting within the two disease groups PDAC and PDAC + DM. The expression of these proteins was normalized to samples of healthy tissue from the same patient. Since the normalization is based on the total protein, no housekeeping proteins are presented.

For the validation based on the Western blot results, three out of four previously identified protein targets by proteomics were verified as differentially expressed in PDAC + DM vs. PDAC tissue samples. The results of the validation of the expression of the remaining proteins are shown in Figure 4 (The original Western blots have been shown in Appendix A).

The examination of the protein CD166 (Figure 4A) showed several bands due to alternative splicing at approximately 205 kDa, 100 kDa, 60 kDa, and 40 kDa, which were differently pronounced. The expression of CD166 was particularly strong in the healthy samples with PDAC and PDAC + DM (80 N, 69 N, 79 N). This was reflected in the lower values for CD166 expression for the diseased tissue of the PDAC + DM groups. Comparing the diseases, the data revealed an increase in the CD166 median between tissue samples of PDAC and PDAC + DM (Figure 4A). Thus, the initial result of the proteomic analysis (logFC: 1.85; Table 3) was confirmed.

TNF-β was not detectable as a single distinct band by Western blot due to its occurrence in multimers (44 kDa, 66 kDa, and 110 kDa) (Figure 4B). After normalization, 43 N stood out with a remarkably elevated level of TNF-β compared to the other samples, with lower expression in the rest of the tissue. This was also reflected in the expression distribution. In PDAC as well as PDAC + DM, TNF-β expression was more or less equal in healthy and diseased tissue. Between the PDAC and PDAC + DM groups, there was no difference in the median expression of TNF-ß. These data were not in line with the previous results of the proteomics analysis, where a slight downregulation of TNF-ß (logFC: −1.23; Table 3) was observed.

Figure 4C shows the expression of CD63 in the individual samples. The extraordinarily strong bands in the PDAC disease group (15 P, 43 P, 82 P, 105 N, 105 P) were striking compared to those in the PDAC + DM group. The normalized values also showed a higher proportion of CD63 values in the diseased samples 15 P, 43 P, 82 P, and 105 P. The distribution in the individual disease groups showed a strong increase in CD63 expression in the diseased tissues compared to the healthy tissues except sample 95. By comparing the disease groups beneath each other, it was apparent that CD63 median expression is significantly decreased between tissue samples of patients with PDAC compared to PDAC + DM. This result was contradictory to the previous results of the proteomics analysis, where CD63 showed enhanced expression in the PDAC + DM samples compared to PDAC only (log FC: 1.47; Table 3).

Figure 4D illustrates the expression of S100A13 in the selected tissue samples. A particularly high protein concentration was observed in samples 79 P, 95 N, 96 P, and 41 P of the PDAC + DM groups. On the other hand, low expression of S100A13 was detected in the 15 N/15 P samples from the PDAC group. For the expression inside the groups, the expression of S100A13 increased for PDAC as well as for PDAC + DM compared to their corresponding normal tissue, except for sample 95. When comparing the different disease groups, a decrease in the median expression of S100A13 between samples with PDAC and PDAC + DM was visible (Figure 4D). Hence, the results of the previously conducted proteomics study (logFC: −1.75; Table 3) were confirmed.

In the Western blots, CDH-1 was detected in three bands (Figure 5A). In general, the expression of CDH-1 was highly heterogenic. In PDAC + DM samples, the normalized data showed a comparably low expression of CDH-1 in the diseased tissues. Between the different disease groups, no notable change in median expression from values of the PDAC group compared to the PDAC + DM group was apparent.

Changes in p53 expression are shown in Figure 5B. In the Western blot, the samples 80 N, 79 N, 56 N, 88 N, and 88 P stand out, in which the detected bands were feeble. Even after normalization, these samples showed an extremely low expression of p53. In the case of the normalized values, sample 15 P was apparent with a high expression of p53. An increased p53 expression in the diseased tissue compared to the healthy tissue was determined. Comparing the diseased groups, median p53 expression was slightly increased between PDAC and PDAC + DM tissues (Figure 5B).

The expression of c-Jun in the different disease groups was already observed by Western blotting (Figure 5C). The amount of protein in PDAC in healthy and diseased tissue was largely constant in strength (15 N, 15 P, 43 N, 43 P), while in PDAC + DM, there was a clear decrease in the amount of protein (69 N, 69 P, 79 N, 79 P, 95 N, 95 P). The normalized values confirm these results with a particularly high expression of c-Jun in the 80 N and 105 N samples. Already apparent from the Western blot, the expression compared in the box plot of the diverse groups showed an evident decrease in median c-Jun expression between PDAC and PDAC + DM.

The analysis of the expression of Smad4 is illustrated in Figure 5D. The detected bands were uniform and showed a reduced band thickness only in the healthy tissue samples of patients 80 and 79 compared to the diseased samples. Sample 15 P stood out with an extraordinarily strong expression of Smad4. The expression in PDAC and PDAC + DM increased in the diseased tissue in both groups compared to their corresponding healthy tissue. Regarding the diverse groups, median expression values showed no significant differences between the two entities.

In the Western blot for TGM2, an increase in protein expression in the diseased samples in contrast to the healthy samples was detected for both the PDAC and PDAC + DM groups (Figure 5E). In particular, the expression in diseased samples in PDAC + DM stood out with high expression. This was observed in the normalized data except for samples 15 and 105, as well. Comparing both groups, the boxplot reflected this increase in TGM2 median expression in the diseased tissue in PDAC + DM vs. PDAC.

The expression of YAP in the selected samples is shown in Figure 5F. Particularly high protein levels were detected in the 80 N and 80 P (PDAC), 69 N, 69 P, and 79 N (PDAC + DM) samples. When comparing the samples within the groups, no altered expression in healthy and diseased tissues was detectable for PDAC or PDAC + DM. There was a substantially lower expression between the median values of YAP in the PDAC and PDAC + DM groups.

The differential expression of miR-1285, initially detected by the miRNA array, was confirmed via qRT-PCR investigation. miR-1285-5p proved to be slightly downregulated, and miR-1285-3p was strongly downregulated in PDAC+ DM compared to PDAC. Concerning target proteins of miR-1285, four target proteins of miR-1285, c-Jun, p53, Smad4, and TGM2 appeared to be differentially expressed between PDAC and PDAC + DM. p53 and CDH-1 were not found to be differentially regulated between these entities. The respective differential regulation of the protein candidates from the proteomics analysis could not be entirely confirmed. The Western blots showed contradictory results to the proteomic analysis in the case of CD63 and TNF-ß. Whereas the expression of CD63 in the proteomics analysis increased between samples of PDAC and PDAC + DM (LogF 1.47) (Table 3), the Western blot results showed a decrease in median protein expression of 32%. TNF-ß expression values were found to be slightly downregulated between the PDAC and PDAC + DM samples during proteomics (LogF −1.23) (Table 3), while validation via Western blotting revealed no significant changes between these two entities. However, of the initial candidates, CD166 and S100A13 showed a consistent differential expression between the PDAC and PDAC + DM disease groups in both experimental settings.

## 4. Discussion

Early detection of PDAC remains a great challenge. Current screening techniques require invasive and expensive tests, and the conventional CA 19-9 marker is not reliable in diagnosing PDAC at an early stage of development [25,33,34,35]. In addition, there is a lack of diagnostic markers to detect DM as an early PDAC symptom [24]. Herein, we attempted to uncover biomarkers for patients suffering from PDAC + DM. Within this study, important biomarkers concerning patients with PDAC + DM compared to PDAC were identified with the differentially expressed proteins CD166, S100A13, c-Jun, YAP, p53, and TGM2 as well as the miRNA-1285-3p. To avoid differences in individual genetic levels and differences in blood samples affecting the accuracy of the results, miRNA and protein expression were normalized to healthy pancreatic tissue of individual patients.

The type and course of diabetes have an important impact on the risk of pancreatic cancer appearance. Diabetes mellitus (DM) includes several types of metabolic disorders characterized by a chronic increase in glucose levels in the blood. The two main forms are medically differentiated: autoimmune disease type 1 DM (T1DM) and T2DM, which occurs more frequently with increasing age. In addition to the two main forms, it is possible to develop gestational diabetes (type 4) during pregnancy or type 3 DM (T3DM) because of an acquired or congenital disease, with DM resulting from pancreatic disease being classified as type 3c (T3cDM). As part of this work, T2DM and T3cDM are examined in more detail.

About 90% of diagnosed diabetics have T2DM. This type occurs in adults and is, therefore, also called adult-onset diabetes. One of the first steps towards diabetes is the inhibition of IRS, PI3K, or Akt in the PI3K-Akt signaling pathway, which leads to insulin resistance, although the exact process leading to the development of T2DM has not yet been fully elucidated. If the signaling pathway is disrupted at one of these points, the transmission of the insulin signal is interrupted, and the translocation of GLUT-4 to the membrane does not take place. As a result, hyperglycemia develops, which is characteristic of DM, and which leads to insulin overproduction by the β-cells to the point of failure. Hyperglycemia can promote the development of a tumor due to the locally high glucose concentrations in the tumor microenvironment (TME) [36].

Hyperinsulinemia because of insulin overproduction leads to increased binding of insulin to the IR and to the non-preferred IR/IGFR complexes, which leads to an excessive and uncontrolled influence on growth, proliferation, and differentiation. The result is atherosclerosis and inflammatory processes, which can also contribute to the development of a tumor [37,38].

Pancreatic diabetes, T3cDM, occurs secondary to pancreatic diseases such as chronic pancreatitis, hemochromatosis, cystic fibrosis, or pancreatic carcinoma. What all forms of T3cDM have in common is that the exocrine function of the pancreas is impaired.

However, the diagnosis of T3cDM is often problematic because T2DM itself leads to abnormalities in the pancreas, and T3cDM is often misdiagnosed due to similar diagnostic criteria. In the case of pancreatic cancer, however, there is an additional problem. On the one hand, T2DM is a risk factor for the development of PDAC and is, therefore, a cause, and on the other hand, PDAC is often only recognized extremely late due to the ambiguous symptoms in the initial stages of the disease and the lack of diagnostic markers. Furthermore, it has been shown that T3cDM, as a symptom of PDAC like T2DM, also has insulin resistance, which is not the case with other T3cDM. This makes it possible that new-onset DM in old age is often diagnosed as T2DM and is not recognized as an early symptom and secondary disease of PDAC [24,39].

Epidemiological studies show that in 70% of PDAC cases with diabetes, PDAC was diagnosed shortly after DM diagnosis or up to 24 months after diagnosis. The risk of developing PDAC in new-onset DM is four to seven times higher in the first two years and decreases the longer the diabetes has been present. A PDAC diagnosis within the first three years of DM diagnosis could indicate T3cDM. After approximately three years of DM, it is a long-term DM that is classified as a risk factor for PDAC [21]. However, diagnostic markers are still missing to recognize new-onset DM as an early symptom of PDAC and to differentiate it from T2DM.

The proteomics analysis of the Sciomics Research Institute in Heidelberg identified proteins that were differentially expressed in PDAC + DM compared to PDAC tissue. The differential expression of CD166, CD63, S100A13, and TNF-β in PDAC vs. PDAC + DM tissue was validated using Western blot analysis. Two proteins, CD166 and S100A3, were successfully verified as differentially expressed between tissue samples of both entities.

Cluster of differentiation (CD) 166, also known as activated leukocyte adhesion molecule (ALCAM), is a cell-surface localized glycoprotein that modulates cell–cell adhesions [40,41]. It is present in different isoforms due to alternative splicing and can be separated at various locations on its extracellular domain (ECD) after localization on the membrane. This creates soluble CD166 fragments of different lengths [42]. CD166 is a well-known prognostic marker for diverse types of cancer. Depending on the type of cancer, an increased or decreased concentration, and localization in the cytoplasm or on the membrane can indicate a poor prognosis. CD166 was detected on the surface of circulating tumor cells (CTCs) [43]. Additionally, in PDAC patients with a KRAS driver mutation, it is hypothesized that CTCs expressing CD44/CD166 on their surface are associated with more rapid lymph node invasion and metastasis [43]. For PDAC, CD166 is already known as a marker, and elevated levels of cytoplasmic CD166 have been associated with tumor recurrence and poor overall survival (OS) [44]. The results of the proteomics study, as well as the results of the following Western blot analysis, show that CD166 expression is upregulated in PDAC + DM compared to PDAC. Therefore, these results are in line with the previous data. In other studies, CD166 increased in the serum, as well [45]. CD166 has already been identified as a possible biomarker for diabetic nephropathies in patients with DM due to increased serum concentrations [41]. Our data reflect both studies by showing a further upregulation in patients with PDAC and additional DM in tissue samples.

S100A13 is a calcium-binding protein of the S100 protein family with a tissue-specific expression. S100A13 expression is upregulated in several tumor entities, such as non-small cell lung cancer (NSCLC), melanoma, and astrocytoma, and correlates with tumor angiogenesis [46,47,48]. Transcriptomics and proteomics studies on tissue-specific expression of S100A13 show low levels of expression in healthy pancreatic tissue [49]. In contrast, increased S100A13 expression in PDAC was detected via mRNA expression analysis in patients with PDAC, although not classified as differential [50]. When examining the methylation status in chronic T2DM with and without diabetic retinopathy compared to T2DM with shorter disease duration, differential methylation of S100A13 was found in diabetic retinopathy, and S100A13 was identified as a possible biomarker [51]. Concerning S100A13 expression in this study, the results of the Western blots comply with the results of the proteomics study. In the proteomics analysis, S100A13 was downregulated in PDAC + DM samples compared to PDAC samples. A decreased protein level in PDAC+ DM compared to PDAC was also detected via Western blots, which indicates a lower influence of S100A13 on tumor angiogenesis in these PDAC patients, rendering S100A13 an interesting marker for patients with PDAC+ DM.

Cluster of differentiation (CD) 63, as well as CD166, is localized on the cell surface and leads to activation of the phosphoinositide 3-kinase (PI3K)/Akt, mitogen-activated protein kinase (MAPK), and focal adhesion kinase 1 (FAK) signaling pathways [52] via complexation with its receptor tissue inhibitor of metalloproteases-1 (TIMP-1) and integrin-β. The proteomics results of the Sciomics Institute showed differential upregulation of CD63 expression in PDAC and DM compared to PDAC patients. In contrast, Western blot analyses detected a slight downregulation of CD63 in PDAC and DM. The upregulation in patients with only PDAC is consistent with results from Kushman et al., who detected elevated CD63 levels in primary tumors and suggested a correlation between CD63 elevation in PDAC and improved overall survival (OS) and progression-free survival (PFS) [53]. Thus, the question arises as to whether the downregulation of CD63 translates into worse OS and PFS for PDAC patients with concomitant DM disease.

TNF-β is a cytokine involved in inflammatory processes and lymphangiogenesis. In PDAC, TNF-β is generally expressed at low levels [54], and decreased TNF-β expression has even been demonstrated in advanced PDAC [16]. According to the Sciomics Institute studies, TNF-β is downregulated in PDAC and DM compared to PDAC patients. This was confirmed via Western blot analysis, where, in turn, a slight downregulation in TNF-β protein expression was detected in the diseased tissue of PDAC+ DM compared to tissue samples in PDAC patients. Concerning cytokines, a differential downregulation in autoimmune pancreatitis has already been shown, which allows differentiation between acute and chronic pancreatitis [55]. Based on the further downregulation in the diseased tissue of the PDAC+ DM group, TNF-β may be a marker for PDAC with diabetes.

MicroRNAs (miRNAs/miRs) are approximately 20–22 nucleotide-long noncoding RNA fragments that can regulate expression at the transcriptional and translational levels by binding to both DNA and mRNA [56]. In this way, they can also influence tumor progression and carcinogenesis by acting either as onco-miRNAs or tumor suppressor miRNAs [57,58,59,60,61]. During miRNA processing, a 3p and/or 5p miRNA can be formed from the 3′ and 5′ strands of the precursor molecule. Various studies have shown that miRNAs are stable in serum/plasma [62] and, hence, can be detected by various assays, such as miRNA microarrays and qRT-PCR [63]. More importantly, unique serum/plasma miRNA expression profiles for PDAC can serve as fingerprints for their detection [16]. The miRNA array performed in collaboration with the Integrated Biobank of Luxemburg uncovered miR-1285 as differentially downregulated in PDAC + DM vs. PDAC patients using plasma blood samples. In the validation step, the regulation of the miR-1285-5p and miR-1285-3p was measured in serum samples from patients with resected PDAC + DM vs. PDAC. A slight downregulation of miR-1285-5p in PDAC + DM compared to PDAC and a strong downregulation of miR-1283-3p in PDAC + DM were observed. This confirms the differential regulation identified in the preliminary miRNA array results. The miR-16-5p used for normalization in the current study is stable in both disease groups (PDAC, PDAC + DM) and can be recommended for normalization. This study reveals miRNA-1285-3p as a novel biomarker in patients with PDAC + DM, and to our knowledge, it has not been previously described in this context in the literature.

miR-1285 has been identified in different types of cancer and several target proteins have been described in the literature, through which it promotes or suppresses tumor progression [26,27,28,29,30] (Table 4). To date, miR-1285 has only been described in PDAC by Huang et al., who also identified YAP as a target protein of miR-1285 in PDAC [30].

Yes-associated protein (YAP), also known as YAP1 or YAP65, is a transcriptional coactivator in the Hippo signaling pathway that can influence cell proliferation and migration and determine organ growth via the TEA domain (TEAD) family transcription factors. When inactivated, it leads to the restriction of cell proliferation and tissue growth [64,65,66]. In pancreatic cancers, YAP expression is upregulated, and overexpression correlates with the occurrence of liver metastases and poor prognosis [67]. In PDAC with KRAS driver mutations, activated YAP promotes tumorigenesis, metastasis, chemoresistance, and metabolic homeostasis and regulates immunosuppressive TME [68,69,70]. Negative regulation of YAP by miR-1285-3p has been shown in the pancreatic carcinoma cell lines T3M4 and SU.86.86, and experiments in osteosarcoma cell lines have confirmed this [30,31]. YAP activation by elevated glucose levels has been demonstrated in vitro and in vessels from db/db mice. YAP has also been identified as a regulator of monocyte attachment and endothelial inflammation induced by high glucose levels [71].

In contrast, in our expression study, in patients with PDAC + DM a decreased expression compared to PDAC patients was present. Negative regulation of YAP by miR-1285-3p as a therapeutic approach in PDAC has been previously presented in T3M4 and SU.86.86 cell lines [30]. Since both YAP and miRNA were downregulated in PDAC and DM, negative regulation of the protein by miR-1285-3p directly in the diseased tissue, as well as a therapeutic approach in the PDAC + DM group, is unlikely.

Cellular Jun (c-Jun) is the product of the proto-oncogene Jun and is active in dimeric form as a transcription factor activating protein 1 (activating protein 1, AP1), which is responsible for the proliferation, differentiation, apoptosis, and oncogenic transformation control [72,73]. In PDAC, the subtypes can be differentiated via AP1. In the classical subtype, which is more responsive to chemotherapy, transcriptional control occurs via JunB/AP1, while in the more aggressive and chemoresistant basal-like subtype, transcriptional control occurs via c-Jun/AP1. Activation of the c-Jun/C-C motif chemokine ligand 2 (CCL2) signaling axis in the classic subtype recruits TNF-α-producing macrophages into the EMT and changes the subtype to a basal-like subtype. In hepatocellular carcinoma (HCC), Jun is a direct target of miR-1285-3p, which can inhibit the proliferation of HCC cells by repressing Jun expression. miR-1285-3p is significantly downregulated in patients who have responded poorly to transarterial chemoembolization (TACE) compared to HCC patients with a good therapeutic response [27]. Serna et al. showed in mice that hepatitis C virus structural protein NS5A-induced c-Jun expression leads to HCC development with diabetes and that c-Jun overexpression inhibits the expression of a rapamycin-insensitive companion of mTOR (RICTOR), which means that mTOR complex 2 (mTORC2) cannot be formed [74]. Due to the lack of activation of Akt by mTORC2, this leads to an inhibition of the PI3K-Akt signaling pathway and results in the expression of genes for gluconeogenesis and self-renewal of tumor-induced stem cell-like cells. An influence of c-Jun on metabolism has been shown in HCC, where the protein is activated via the GLUT-2 promoter, which inhibits GLUT-2 expression, preventing glucose uptake [74]. Our expression studies on c-Jun show a constant expression in PDAC but strongly downregulated median expression between diseased tissue of PDAC vs. PDAC + DM. However, since both c-Jun and miR-1285-3p are strongly downregulated, regulation of c-Jun expression by miRNA in the PDAC + DM disease group is unlikely.

p53 is a tumor suppressor that, after activation by stress, acts as a transcription factor and induces the expression of genes for cell-cycle regulation, senescence, or apoptosis [75]. As one of the driver mutations in PDAC, the tumor suppressor p53 is affected by mutations in 50–75% of PDAC diseases, which can lead to either loss or gain of function and accordingly differentially affect the tumor [75,76]. It is possible to modulate antigen presentation, regulate the expression of activating ligands for natural killer (NK) cells and immunosuppressive molecules, and influence cytokine and chemokine secretion by mutated p53 [77]. In addition to regulating the cell cycle and apoptosis, p53 also influences the metabolism and the development of DM by promoting insulin resistance through the regulation of glucose uptake via GLUT-1, negative regulation of glycolysis, and positive regulation of gluconeogenesis, thus increasing blood glucose levels [78]. For the samples studied here, increased median p53 expression was detected in the diseased tissue of PDAC+ DM compared to the expression in the diseased tissue of PDAC patients. It is possible that p53 affects metabolism by promoting insulin resistance through regulation of glucose uptake via GLUT-1, negative regulation of glycolysis, and positive regulation of gluconeogenesis, leading to increased glucose levels in the blood [78]. Therefore, the increased p53 expression in PDAC + DM leaves room to question whether it induces or enhances the formation of DM in PDAC.

Exactly how p53 is regulated remains an area of uncertainty, but in neuroblastoma, hepatoblastoma, and breast cancer, p53 has been identified as a target protein of miR-1285-3p [28]. If miR-1285-3p in PDAC acts cancerogenically and represses p53 expression, this could explain the increased p53 expression in this group by downregulation of miR-1285-3p in PDAC + DM. However, as p53 is often mutated in PDAC, the tumor suppressor is rather unsuitable as a differential biomarker.

The cell adhesion protein CDH-1 localized on the cell surface is involved in tissue organization and can thus prevent tumor invasion and metastasis [79,80]. Siret et al. demonstrated decreased CDH-1 expression in PDAC but still detected CDH-1 on the cell surface. In PDAC, cooperation with cadherin-3 prevents CDH-1 from negatively affecting tumor growth and positively enhances tumor aggressiveness and invasiveness [81]. In non-small cell lung cancer (NSCLC), CDH-1 is downregulated by the oncogenic miR-1285-5p [26]. Since miR-1285-5p is slightly downregulated in PDAC + DM and CDH-1 is upregulated in diseased PDAC + DM samples compared to healthy samples, regulation of CDH-1 by miR-1285-5p in patient samples of PDAC+ DM would likely be based on these results. This renders CDH-1 a possibly suitable biomarker for PDAC+ DM.

Western blot analysis revealed a slight but not significant upregulation of Smad4 (Mothers against decapentaplegic homolog 4) expression in samples from patients with PDAC+ DM compared to patients with PDAC. Similar to p53, Smad4 is a driver mutation in PDAC whose inactivating mutations lead to loss of regulation of transforming growth factor-β (TGF-β) and have protumorigenic effects [76]. In podocytes, Smad4 deficiency has also been shown to affect metabolism, leading to increased glycolysis and lactate formation. In PDAC, lactate enhances invasiveness, and due to hypoxic conditions, tumors, including PDAC, produce increased lactate, which promotes metastasis and tumor cell viability [82]. As Smad4 is normally inactivated in PDAC and Smad4 deficiency promotes lactate formation, a slight upregulation of the tumor suppressor PDAC + DM points to presumably lower invasiveness and metastasis of the tumor in patients with PDAC+ DM. It is important to mention that that a Smad4 driver mutation is present in only 31–38% of PDAC diseases. The PANTOR project biodatabase is quite small, and the mutation profile of patients in the biodatabase is unknown. There is also the possibility that the selected patients are not affected by mutations in this gene.

Human transglutaminase 2 (TGM2) is a scaffolding protein that mediates the formation of protein complexes and can bind and hydrolyze GTP. Intracellular TGM2 plays a role in the regulation of apoptosis. Extracellular TGM2 serves to stabilize the extracellular matrix and has been linked to cell invasion [83]. Protein expression analysis showed overexpression of TGM2 in IPMN. Genetic analyses identified TGM2 as one of the most strongly expressed genes in PDAC [84,85]. The expression of TGM2 is associated with metastasis via the lymph nodes, lymphovascular invasion, and chemoresistance [86]. A negative regulation of expression by miR-1285 was found in renal cell carcinoma [29]. However, TGM2 was highly upregulated in PDAC tissue compared to healthy tissue and in PDAC + DM compared to healthy tissue. Additionally, there was an evident increase in median TGM2 expression between PDAC and PDAC + DM. miR-1285-3p is differentially downregulated in PDAC + DM and is incapable of suppressing TGM2 expression. Therefore, TGM2 appears to be indirectly regulated by the absence of miR-1285 in PDAC + DM. These findings, especially in view of the role of TGM2, suggest that TGM2 is a promising new potential biomarker for PDAC+ DM.

Limitations and perspectives: Our project represents a pilot approach due to the small number of collected biospecimens and the heterogeneity of DM disease subtypes that could not be further investigated. Due to the study design, it was not possible to set up a control group of diabetic patients without PDAC, as only blood samples could be obtained from this subgroup of people, thus not being able to demonstrate the expression of the screening indicators in this group of people. If this group could be added and investigated, the experimental design would be more rigorous. Although we showed the feasibility of the miRNA and proteomics analysis, the investigation on screening samples should be repeated in larger patient populations to ensure the validity of the findings. Especially in a broader study population, it would be more feasible to illustrate the impact of the type and course of diabetes on the risk of pancreatic cancer appearance.

## 5. Conclusions

Despite the above limitations, the potential biomarkers presented in the current study, the differentially expressed proteins CD166 and S100A13, as well as the miRNA miRNA-1285-3p and their target proteins TGM2 and CDH-1, represent a potential major advance in the search for biomarkers for the diagnosis of PDAC enabling early detection of PDAC in DM patients in the future.

## Figures and Tables

**Figure 1 cancers-16-02726-f001:**
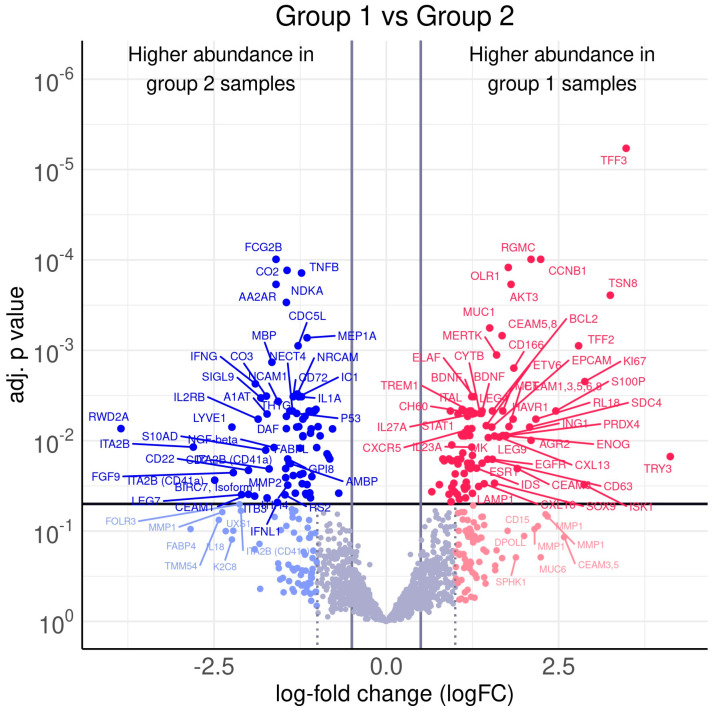
Volcano plot of differences in Protein abundance between PDAC + DM (group 1) and PDAC (group 2): Several proteins exhibited distinct abundance variations in group 1 samples and group 2 samples. The volcano plot visualizes the *p*-values (adjusted for multiple testing) and corresponding log-fold changes (logFC). A significance level of adj. *p*-value = 0.05 is indicated as a horizontal red line. The logFC cutoffs are indicated as vertical lines. Proteins with a positive logFC had a higher abundance in group 1 samples (PDAC + DM) and proteins with a negative value in group 2 samples (PDAC).

**Figure 2 cancers-16-02726-f002:**
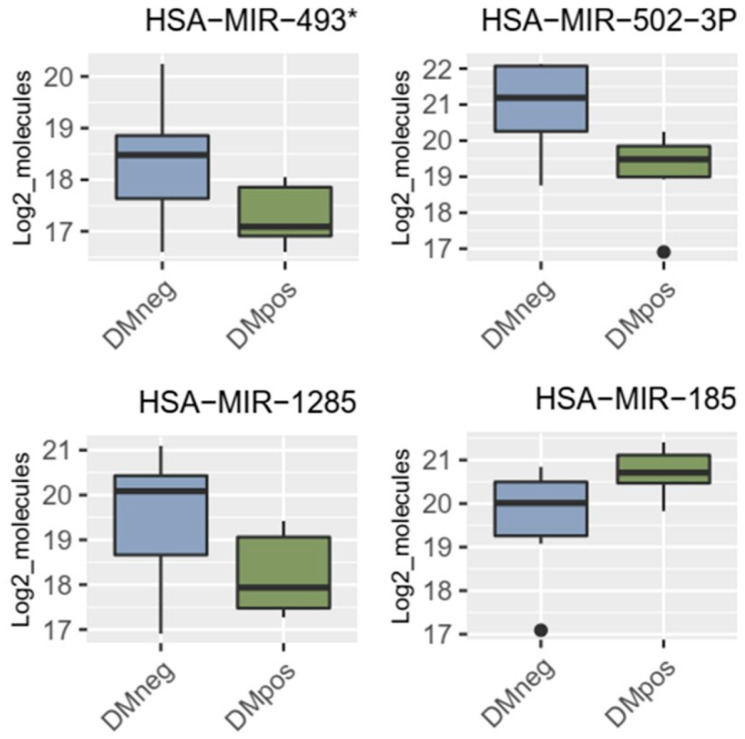
Significantly regulated miRNAs: miRNA-1285, miRNA-502-3p, miRNA-493*, and miRNA-185 in patients with PDAC vs. PDAC + DM in serum.

**Figure 3 cancers-16-02726-f003:**
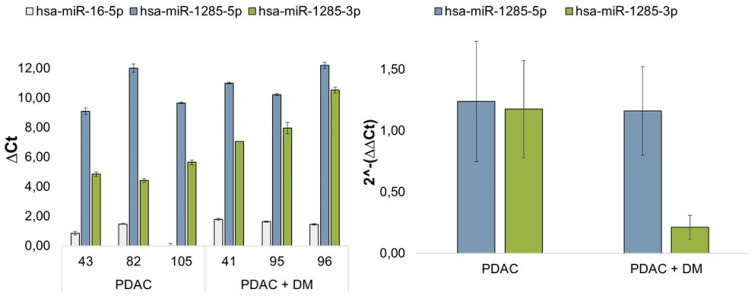
Validation of miR-1285 in patients with PDAC and PDAC + DM: Relative change in miR-1285-3p and miR-1285-5p expression in tissue samples from patients with PDAC and PDAC + DM. Regulation of miR-1285-5p and miR-1285-3p in patients with PDAC + DM compared with patients with PDAC. Normalization was performed via miR-16-5p. Representation of the ΔCt values of miR-16-5p, miR-1285-5p, and miR-1285-5p in the respective patient samples normalized to the respective miR-16-5p Ct values of the sample (miR1285-3p fold change between PDAC and PDAC + DM = 0.18).

**Figure 4 cancers-16-02726-f004:**
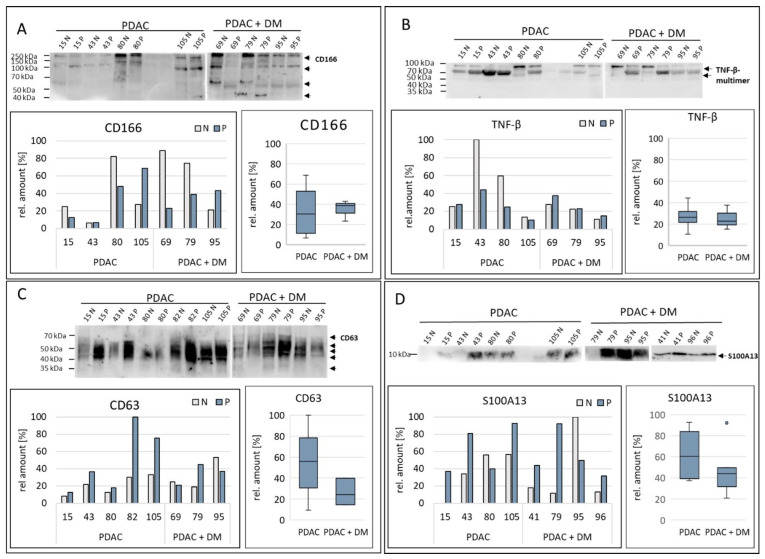
Expression of the differential protein markers that were identified as promising protein targets by the proteomics analysis, CD166 (**A**), TNF-β (**B**), CD63 (**C**), and S100A13 (**D**) in tissue samples from patients with PDAC, or PDAC + DM normalized to total protein. Data shown for each protein: Western blot for protein expression, expression of the protein of interest in the individual samples, and protein expression distribution via box plots in the individual groups normalized to the corresponding healthy tissue of each patient. N: healthy tissue, P: tumor tissue.

**Figure 5 cancers-16-02726-f005:**
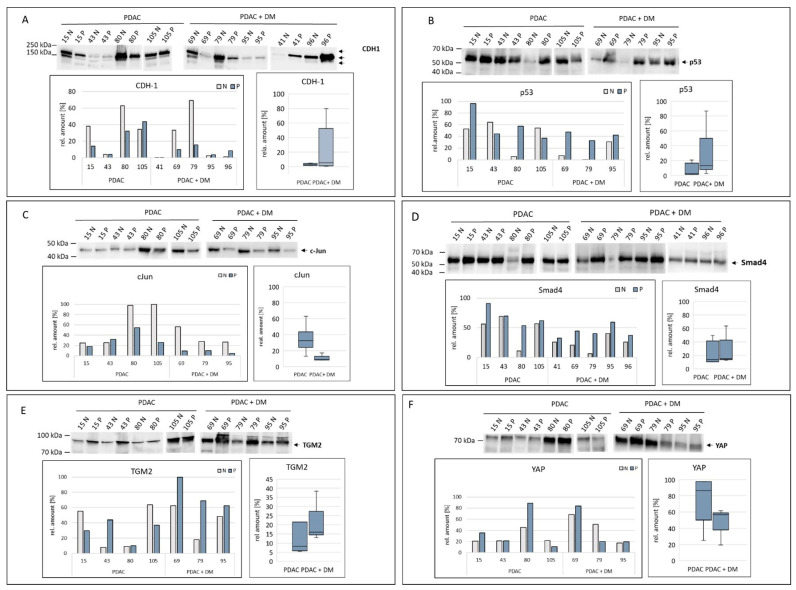
Expression of the target proteins of the differentially expressed miRNA-1285: CDH-1 (**A**), p53 (**B**), c-Jun (**C**), Smad4 (**D**), TGM2 (**E**), and YAP (**F**) were analyzed via Western blotting in tissue samples within the two disease groups PDAC, PDAC + DM normalized to total protein. Data shown for each protein: Western blot for protein expression, expression of the protein of interest in the individual samples, and protein expression distribution via box plots in the individual groups normalized to the corresponding healthy tissue of each patient. N: healthy tissue, P: tumor tissue.

**Table 1 cancers-16-02726-t001:** Clinical and pathology data of patients submitted to pancreatic surgery in UKS from 1 September 2016 to 31 December 2017 (44 patients).

Sample	Age	Gender	BMI	Histology	DM	HbA1C	CA 19-9
43	78	m	32.7	PDAC (no resection)	y, insulin-dependent	11.3	10,974
68	77	m	29.7	PDAC (no resection)	y (not known pre-OP)	6.9	719
35	69	f	25.2	PDAC (no resection)	y, insulin-dependent	9.8	1248
87	80	f	26.6	PDAC (no resection)	y, insulin-dependent	8.1	1803
66	79	m	24.3	PDAC (no resection)	n	5.6	726.8
40	59	m	27.7	PDAC (no resection)	y, insulin-independent	7.3	1294
42	73	m	22.4	PDAC (no resection)	n	5.7	599.6
36	61	f	25.1	PDAC (no resection)	n	5.8	601.8
8	76	m	22.6	PDAC (resected)	n	4.7	256.6
78	81	m	36.4	PDAC (resected)	n	5.1	84.3
82	82	m	22	PDAC (resected)	n	5.1	118.9
71	73	m	28.7	PDAC (resected)	y, insulin-dependent	6.2	189.4
79	70	m	32.3	PDAC (resected)	y, insulin-dependent	9.7	3340
69	64	m	31.2	PDAC (resected)	y, insulin-dependent	7.6	1147
44	62	m	27.8	PDAC (resected)	n	5.4	17.3
80	82	f	27.3	PDAC (resected)	n	5.9	247.3
95	69	m	20	PDAC (resected)	y, insulin-dependent	6.2	0.6
105	73	f	19.3	PDAC (resected)	n	6.1	368.3
99	80	f	20.3	PDAC (resected)	n	6	1289
51	63	f	20.2	PDAC (resected)	n	4.9	335.5
62	75	f	23.4	PDAC (resected)	n	5.2	24.9
80	82	f	29.4	PDAC (resected)	n	4.1	3917
15	78	f	25.6	PDAC (resected)	n	6.2	208.6
41	75	f	27.6	PDAC (resected)	y, insulin-independent	7.6	127.4
86	58	f	27.9	PDAC (resected)	n	6	218.4
96	55	f	36.1	PDAC (resected)	y, insulin-independent	5.3	1
43	78	f	23	PDAC (resected)	n	5.9	211.3
29	61	m	24.5	chronic pancreatitis	n	5.6	9.5
1	57	m	32	chronic pancreatitis	y, insulin-dependent	7.6	11
100	35	m	29.6	chronic pancreatitis	y, insulin-dependent	9.9	9.8
89	61	m	26.8	chronic pancreatitis	y, insulin-independent	7.6	5.2
88	44	f	22.1	chronic pancreatitis	y, insulin-independent	6.6	45.5
56	45	f	13.9	chronic pancreatitis	n	4.7	17
76	69	f	27.5	chronic pancreatitis	y, insulin-independent	5.7	-
52	33	f	22	chronic pancreatitis	n	5.6	<0.6
67	78	f	27.1	chronic pancreatitis	n	6.4	9.3
9	71	f	25.3	chronic pancreatitis	y, insulin-dependent	7.6	20.6
85	65	m	27.7	distal CCC	y, insulin-independent	6.9	32.1
58	74	m	32.7	distal CCC	n	5.1	64.3
24	81	m	23.9	distal CCC	n	5.4	85.7
73	78	m	27.7	distal CCC	n	5.3	8
53	68	m	23.2	distal CCC	n	5.9	76
93	77	m	24	distal CCC	n	5.6	159
75	74	f	27.7	NET Pancreas	n	5.4	5.8

Gender: m (male), f (female), BMI in kg/m^2^, PDAC: pancreatic ductal adenocarcinoma, DM: diabetes mellitus, y: yes, n: no, HbA1C value in %, CA 19-9 value in, distal CCC: distal cholangiocarcinoma, NET: neuroendocrine tumor.

**Table 2 cancers-16-02726-t002:** Selected KEGG pathways related to proteins with differential abundance in Group 1 vs. Group 2.

Pathway ID		Pathway Description	Protein Count
1	hsa05200	Pathways in cancer	29
2	hsa04060	Cytokine-cytokine receptor interaction	22
3	hsa04151	PI3K-Akt signaling pathway	16
4	hsa05418	Fluid shear stress and atherosclerosis	14
5	hsa04514	Cell adhesion molecules (CAMs)	14
6	hsa05205	Proteoglycans in cancer	13
7	hsa04010	MAPK signaling pathway	13
8	hsa05225	Hepatocellular carcinoma	12
9	hsa04015	Rap1 signaling pathway	12
10	hsa04380	Osteoclast differentiation	11
11	hsa04630	Jak-STAT signaling pathway	11
12	hsa05152	Tuberculosis	11
13	hsa04014	Ras signaling pathway	11
14	hsa04640	Hematopoietic cell lineage	10
15	hsa05161	Hepatitis B	10
16	hsa05226	Gastric cancer	10
17	hsa05223	non-small cell lung cancer	9
18	hsa05218	Melanoma	9
19	hsa04933	AGE-RAGE signaling pathway in diabetic complications	9
20	hsa05162	Measles	9
21	hsa05166	HTLV-I infection	9
22	hsa05165	Human papillomavirus infection	9
23	hsa05214	Glioma	8
24	hsa05212	Pancreatic cancer	8

**Table 3 cancers-16-02726-t003:** Potential differentially expressed proteins in tissue samples from patients with PDAC + DM. Result of a proteomics analysis with tissue samples, both diseased and healthy, from patients with PDAC, PDAC + DM.

Protein	Regulation PDAC + DM	Potential Validation Method
BDNF	↑	WB
CD166	↑	WB
CD63	↑	WB
CXCL13	↑	WB
CXCL16	↑	WB
S100A13	↓	WB
TNF-β	↓	WB

WB: Western Blot; ↑: upregulated; ↓: downregulated.

**Table 4 cancers-16-02726-t004:** Significant miRNAs in patients with DM and resected PDAC tumor.

miRNA	logFC	AveExpr	t	*p*. Value	adj. *p*. Value	B
HSA-MIR-502-3P	1.793496999	20.34749718	3.313711041	0.003136645	0.366987411	−1.957157871
HSA-MIR-493*	1.007480749	17.97002518	2.405485145	0.024933313	0.906929645	−3.328008992
HSA-MIR-1285	1.250109763	19.05366401	2.132340797	0.044305461	0.906929645	−3.704638277
HSA-MIR-185	−0.935114451	20.09011602	−2.089490862	0.048357219	0.906929645	−3.761480577

**Table 5 cancers-16-02726-t005:** Significant miRNAs in patients with DM and unresectable PDAC tumor.

miRNA Symbol	logFC	AveExpr	t	*p*. Value	adj. *p*. Value	B
HSA-MIR-1273	−2.43083379	19.1112214	−2.9404612	0.0132589	0.64242329	−4.55313887
HSA-MIR-1285	−2.57078247	18.9186008	−2.87550303	0.01490307	0.64242329	−4.55444384
HSA-MIR-371-5P	−1.71306615	18.0949013	−2.79817459	0.01712805	0.64242329	−4.55602414
HSA-MIR-888	−1.94922757	18.4567094	−2.78193133	0.01763595	0.64242329	−4.55635971
HSA-MIR-1274B	−1.71617505	23.1619678	−2.71333725	0.01995103	0.64242329	−4.5577903
HSA-MIR-122	−2.18577019	21.34131	−2.55501492	0.02650499	0.68321569	−4.56117141
HSA-MIR-195	2.31466878	21.5462484	2.18958784	0.05068444	0.68321569	−4.56932364
HSA-MIR-192	−1.7910867	18.5855867	−2.16196007	0.05319605	0.68321569	−4.5699547
HSA-MIR-424	1.94472688	20.6816518	2.14824269	0.05448657	0.68321569	−4.57026855
RNU2-1	−1.55939277	20.9494415	−2.12343529	0.05689613	0.68321569	−4.57083695

Only miR-1285 was significantly downregulated in PDAC + DM patients.

**Table 6 cancers-16-02726-t006:** Target proteins of the miRNA-1285-3p and 5p variants and tumor entities in which they have been described.

miR-Variant	Effect	Tumor Entity	Protein Target	Reference
miR1285-3p	Tumor suppressor	HCC	Jun ↓	(Liu et al., 2015 [27])
	Tumor promoter	Neuroblastoma Hepatoblastoma Mamma carcinoma	p53 ↓	(Tian et al., 2010 [28])
	Tumor suppressor	RCC	TGM2 ↓	(Hidaka et al., 2012 [29])
	Tumor suppressor	PDAC OS	YAP ↓	(Huang et al., 2017 [30]) (Hu et al., 2019 [31])
miR-1285-5p	Tumor promoter	NSCLC	CDH-1 ↓ Smad4 ↓	(Zhou et al., 2017 [26])

↑: upregulated; ↓: downregulated.

## Data Availability

The mass spectrometry proteomics data that support the findings of this study have been deposited to the ProteomeXchange Consortium via the PRIDE [1] partner repository with the dataset identifier PXD053169. Our data are available upon reasonable request.

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
