# Peer review of "Differential miRNA and Protein Expression Reveals miR-1285, Its Targets TGM2 and CDH-1, as Well as CD166 and S100A13 as Potential New Biomarkers in Patients with Diabetes Mellitus and Pancreatic Adenocarcinoma"

_cancers, 2024, doi:10.3390/cancers16152726_

Round 1

Reviewer 1 Report

Comments and Suggestions for Authors

The study aimed to examine miRNA and protein biomarkers for early PDAC diagnosis among patients with diabetes mellitus. 

There several questions regarding study design:

1. Authors examined expression miRNAs and Protein among patient samples. However, miRNA expression analysis was performed using plasma while the protein expression analysis was performed on the tissue. Authors then claimed that increased expression of mIR-1285 in serum reduces target protein expression (CDH-1, p53, c-Jun, smad4, TGM2 and YAP) in tissue. There are no evidence that serum miR-1285 is taken by the pancreatic tumor cells or tissue cells. Therefore, it remains non-conclusive that miR-1285 is responsible for reduced target expression in tissue. 

2. Furthermore, the study did not use any housekeeping gene as a control to show equal loading of protein through the study. Which potentially could bias and false interpret the results. 

3. Western blot of CD166 (abcam clone 3D9F1) is expected to show a band at 47 kDa. However, authors show multiple bands and different molecular weight which doesn't seem to correspond with the manufacturer's datasheet. 

4. TNF-beta also is expected to show band at 22kDa. However authors claimed TNF-beta multimers at 50, 70 and 100kDa. 

Comments on the Quality of English Language

The language is good and easy to read. 

Author Response

Thank you for your valuable comments and suggestions regarding our manuscript:  “Differential miRNA and protein expression reveals miR-1285, its targets TGM2 and CDH-1 as well as CD166 and S100A13 as potential new biomarkers in patients with diabetes mellitus and pancreatic adenocarcinoma” for publication in the Journal Cancers.

We appreciate the time and effort that you dedicated providing feedback on our manuscript and are grateful for the insightful comments on and valuable improvements to our paper.

Please see below, in blue, for a point-by-point response to your comments and concerns.

The study aimed to examine miRNA and protein biomarkers for early PDAC diagnosis among patients with diabetes mellitus.

There are several questions regarding study design:

  1. Authors examined expression miRNAs and Protein among patient samples. However, miRNA expression analysis was performed using plasma while the protein expression analysis was performed on the tissue. Authors then claimed that increased expression of mIR-1285 in serum reduces target protein expression (CDH-1, p53, c-Jun, smad4, TGM2 and YAP) in tissue. There is no evidence that serum miR-1285 is taken by the pancreatic tumor cells or tissue cells. Therefore, it remains non-conclusive that miR-1285 is responsible for reduced target expression in tissue.

Thank you for this remark. We are very grateful you brought this topic to our attention. We agree that there is no evidence that serum miR-1285 is taken by the pancreatic tumor cells or tissue cells from the blood. Your comment has made us realize that it is not assuming, of course, that it is generally known that miRNA measured in plasma directly reflect the situation in the tumor, because the tumor is secreting tumor-specific miRNAs into the blood.

We cited the work of Mitchell et al. from 2008, where they show clearly, that miRNAs originating from human prostate cancer xenografts enter the circulation, are readily measured in plasma, and can robustly distinguish xenografted mice from controls. Furthermore, they were able to show that this concept extends to cancer in humans, where serum levels of miR-141 (a miRNA expressed in prostate cancer) can distinguish patients with prostate cancer from healthy controls. This group was the first to establish the measurement of tumor-derived miRNAs in serum or plasma as an important approach for the blood-based detection of human cancer.

Mitchell PS et al. , Parkin RK, Kroh EM, Fritz BR, Wyman SK, Pogosova-Agadjanyan EL, Peterson A, Noteboom J, O'Briant KC, Allen A, Lin DW, Urban N, Drescher CW, Knudsen BS, Stirewalt DL, Gentleman R, Vessella RL, Nelson PS, Martin DB, Tewari M. Circulating microRNAs as stable blood-based markers for cancer detection. Proc Natl Acad Sci U S A. 2008 Jul 29;105(30):10513-8. doi: 10.1073/pnas.0804549105. Epub 2008 Jul 28. PMID: 18663219; PMCID: PMC2492472.

Since this work, series of miRNAs in serum have been identified as potential tools for early diagnosis, prediction of treatment response, and prognosis of patients with PDAC.

A recent work of Matsuzaki et al from 2023 could also show, that  miRNA profiles differ between organs and that circulating miRNomics in plasma represent  these organ- specific cancers.

Matsuzaki J, Kato K, Oono K, Tsuchiya N, Sudo K, Shimomura A, Tamura K, Shiino S, Kinoshita T, Daiko H, Wada T, Katai H, Ochiai H, Kanemitsu Y, Takamaru H, Abe S, Saito Y, Boku N, Kondo S, Ueno H, Okusaka T, Shimada K, Ohe Y, Asakura K, Yoshida Y, Watanabe SI, Asano N, Kawai A, Ohno M, Narita Y, Ishikawa M, Kato T, Fujimoto H, Niida S, Sakamoto H, Takizawa S, Akiba T, Okanohara D, Shiraishi K, Kohno T, Takeshita F, Nakagama H, Ota N, Ochiya T; Project Team for Development and Diagnostic Technology for Detection of miRNA in Body Fluids. Prediction of tissue-of-origin of early-stage cancers using serum miRNomes. JNCI Cancer Spectr. 2023 Jan 3;7(1):pkac080. doi: 10.1093/jncics/pkac080. PMID: 36426871; PMCID: PMC9825310.

We added both paper to our manuscript as literature and complemented the introduction with a section dealing with the study design and this specific question posed.

See the revised text in line 104-107.

This will contribute to a better understanding of the paper and the study design. This literature additionally increases the statement of our data for the reader.

Furthermore, when you are interested, there are several other studies that suggest a crosstalk between exosomal miRNAs in serum and target protein expression in tissue, for example following literatures:

1)    Adem B, Bastos N, Ruivo CF, Sousa-Alves S, Dias C, Vieira PF, Batista IA, Cavadas B, Saur D, Machado JC, Cai D, Melo SA. Exosomes define a local and systemic communication network in healthy pancreas and pancreatic ductal adenocarcinoma. Nat Commun. 2024 Feb 21;15(1):1496. doi: 10.1038/s41467-024-45753-7. PMID: 38383468; PMCID: PMC10881969.

2)    Vicentini C, Calore F, Nigita G, Fadda P, Simbolo M, Sperandio N, Luchini C, Lawlor RT, Croce CM, Corbo V, Fassan M, Scarpa A. Exosomal miRNA signatures of pancreatic lesions. BMC Gastroenterol. 2020 May 6;20(1):137. doi: 10.1186/s12876-020-01287-y. PMID: 32375666; PMCID: PMC7204029.

  1. Furthermore, the study did not use any housekeeping gene as a control to show equal loading of protein through the study. Which potentially could bias and false interpret the results.

Thank you for pointing this out. We agree that we were not using housekeeping genes for normalization. Because nearly all proteins we aimed to analyze in this study show multiple bands on the gel, due to multimers, we decided to use the method of Total Protein Normalization. Therefore, no housekeeping proteins are shown on the blots pictured in the paper.

Total Protein Normalization is a technique, where target protein abundance is normalized to the overall amount of protein present in a sample rather than a housekeeping protein to minimize the impact of varying expression of a loading control. Total protein normalization is used more frequently in western blotting due to the improved reliability of normalized expression values.

You can find additional information here:

https://www.bio-rad.com/de-de/applications-technologies/total-protein-normalization?ID=PODYJQRT8IG9.

The method is described in detail in the material and method section:

“Western blot with normalization over total protein” in line 283-306 of the manuscript.

  1. Western blot of CD166 (abcam clone 3D9F1) is expected to show a band at 47 kDa. However, authors show multiple bands and different molecular weight which does not seem to correspond with the manufacturer's datasheet.

Thank you for your comment. Cluster of differentiation (CD) 166, also known as activated leukocyte adhesion molecule (ALCAM), is a glycoprotein localized on the cell surface to modulate cell-cell adhesion and it is present in different isoforms due to alternative splicing and can be separated at different sites after localization to the membrane at its extracellular domain (ECD). This results in soluble CD166 fragments of different lengths. That is why, especially in tissue, you can observe multiple bands in western blotting.

We added the text: “due to alternative splicing” in line 418-419, so the reader can refer to this.

Find additional information about CD166 isoforms at Hebron et al, 2008:

Hebron, Katie E.; Li, Elizabeth Y.; Arnold Egloff, Shanna A.; Lersner, Ariana K. von; Taylor,

Chase; Houkes, Joep et al. (2018): Alternative splicing of ALCAM enables tunable regulation

of cell-cell adhesion through differential proteolysis. In: Sci Rep 8 (1), S. 3208. DOI:

10.1038/s41598-018-21467-x

  1. TNF-beta also is expected to show band at 22kDa. However, authors claimed TNF-beta multimers at 50, 70 and 100kDa.

Thank you for pointing this out. There was a mistake in the text. Of course, must be “TNF-β was not detectable as a single distinct band by Western blot due to its occurrence in multimers, (44 kDa, 66 kDa and 110 kDa) (figure 4B).” Find the revised text in line 427.

Comments on the Quality of English Language

The language is good and easy to read.

Thank you!

Reviewer 2 Report

Comments and Suggestions for Authors

The authors reported and identified a few new biomarkers for PDAC+DM, including miR-1285, and their targets TGM2 and CDH-1, using numerous biochemical and proteomic tools. Careful and detailed evaluations at transcriptional and translational levels of those biomarkers supported their conclusions, while the limitations of study scope and more rigorous controls were discussed. In general, the study results are solid. Based on the proteomics (PDAC+DM Vs PDAC), it would be interesting to evaluate any pathways specifically enriched/related in PDAC+DM. Combining with the western and qPCR validations, further evaluations on the role of the newly identified biomarkers in those specific potential pathways could reveal the related mechanism.

Minor:

1.        Text font size in the third/fourth paragraph(line71/75) in the introduction seems smaller than the other parts

2.        FIGURE 1 resolution is not ideal, and should be improved. The new biomarkers (up or downregulated proteins) can be highlighted with different colors for better data visualization/presentation.

3.        In figure 3, the qPCR data is usually presented in the form of fold changes.

Author Response

Thank you for your valuable comments and suggestions regarding our manuscript:  “Differential miRNA and protein expression reveals miR-1285, its targets TGM2 and CDH-1 as well as CD166 and S100A13 as potential new biomarkers in patients with diabetes mellitus and pancreatic adenocarcinoma” for publication in the Journal Cancers.

We appreciate the time and effort that you dedicated providing feedback on our manuscript and are grateful for the insightful comments on and valuable improvements to our paper.

Please see below, in blue, for a point-by-point response to your comments and concerns.

Recommendation for revision:

The authors reported and identified a few new biomarkers for PDAC+DM, including miR-1285, and their targets TGM2 and CDH-1, using numerous biochemical and proteomic tools. Careful and detailed evaluations at transcriptional and translational levels of those biomarkers supported their conclusions, while the limitations of study scope and more rigorous controls were discussed. In general, the study results are solid. Based on the proteomics (PDAC+DM Vs PDAC), it would be interesting to evaluate any pathways specifically enriched/related in PDAC+DM. Combining with the western and qPCR validations, further evaluations on the role of the newly identified biomarkers in those specific potential pathways could reveal the related mechanism.

Minor:

  1. Text font size in the third/fourth paragraph(line71/75) in the introduction seems smaller than the other parts

Thank you for your comment, the text font size was revised (line 100-104, revised manuscript).

  1. FIGURE 1 resolution is not ideal and should be improved. The new biomarkers (up or downregulated proteins) can be highlighted with different colors for better data visualization/presentation.

Thank you for your advice. We reworked figure 1 according to your comment.

  1. In figure 3, the qPCR data is usually presented in the form of fold changes.

Thank you for pointing this out. We chose to present the d-ct of the data, which gives the expression of the analyzed miRNAs in our test samples, because we wanted to show the stable continuous expression of the control miR16-5p in all samples. Then we calculated the ddct values to show the normalized relative expression of miR1285-3p and 1285-5p based on these data. We agree that it will add valuable information to the reader to state the fold change of the miR1285-3p between PDAC and PDAC + DM, so we added this information to the manuscript. See the revised figure description in line 398 and in the information in the text in line 391.

Reviewer 3 Report

Comments and Suggestions for Authors

Recommendation for revision:

Query#1

The introduction was noted to be overly concise and lacking in sufficient detail. Although the authors discussed the aggressiveness of PDAC there are many details that still lack:

1)     I suggest the authors expand the PDAC case history, I don't find it very useful for readers to know only about PDAC cases detected in Germany.

2)     I suggest to the authors to go into detail in the existing therapies for pancreatic cancer reported so far, indeed aside of chemotherapy different targeted therapies based on Immune Checkpoint Inhibitors (ICIs) and small molecule kinase inhibitors (SMKIs) have been reported so far. At this purpose, I suggest to the authors to cite the following updated literatures:

1) Frontiers in oncology, 2021,11, 688377. https://doi.org/10.3389/fonc.2021.688377

2)  Nature medicine, 2016, 22(8), 851–860. https://doi.org/10.1038/nm.4123   

3)  Marine drugs, 21(5), 288. https://doi.org/10.3390/md21050288

4) Marine drugs, 21(7), 412. https://doi.org/10.3390/md21070412  

Query#2

Please improve the section “Results” it lacks a clear description of the data provided in the tables and in the present form, it appears really difficult to follow.

General comment

Double check the entire documents, different typos are present.

Comments on the Quality of English Language

Review the English form of the whole article different parts need careful checking of the English form and language.

Author Response

Thank you for your valuable comments and suggestions regarding our manuscript:   “Differential miRNA and protein expression reveals miR-1285, its targets TGM2 and CDH-1 as well as CD166 and S100A13 as potential new biomarkers in patients with diabetes mellitus and pancreatic adenocarcinoma” for publication in the Journal Cancers.

We appreciate the time and effort that you dedicated providing feedback on our manuscript and are grateful for the insightful comments on and valuable improvements to our paper.

Please see below, in blue, for a point-by-point response to your comments and concerns.

Recommendation for revision:

Query#1

The introduction was noted to be overly concise and lacking in sufficient detail. Although the authors discussed the aggressiveness of PDAC there are many details that still lack:

1)     I suggest the authors expand the PDAC case history, I don't find it very useful for readers to know only about PDAC cases detected in Germany.

2)     I suggest to the authors to go into detail in the existing therapies for pancreatic cancer reported so far, indeed aside of chemotherapy different targeted therapies based on Immune Checkpoint Inhibitors (ICIs) and small molecule kinase inhibitors (SMKIs) have been reported so far. At this purpose, I suggest to the authors to cite the following updated literatures:

1) Frontiers in oncology, 2021,11, 688377. https://doi.org/10.3389/fonc.2021.688377

2)  Nature medicine, 2016, 22(8), 851–860. https://doi.org/10.1038/nm.4123  

3)  Marine drugs, 21(5), 288. https://doi.org/10.3390/md21050288

4) Marine drugs, 21(7), 412. https://doi.org/10.3390/md21070412 

Thank you very much for your suggestions concerning the “Introduction.” We added the following text according to your comments and literature: Lines 60-85: “Similar five-year survival…against PDAC [9]”.

Query#2

Please improve the section "Results" it lacks a clear description of the data provided in the tables and in the present form, it appears really difficult to follow.

Thank you for your advice. We added the following text to the results section to increase the readability and understanding of the reader : Lines 320-332: “Eight patients were… registered in table 1. „, Lines 360-361: „Interestingly… in table 5. „, Lines 373-374: „as it was significantly expressed in both subgroups of patients with resected and unresectable PDAC.“

General comment

Double check the entire documents, different typos are present.

Thank you very much for pointing this out. We overworked the manuscript completely, regarding typing errors.

Comments on the Quality of English Language

Review the English form of the whole article different parts need careful checking of the English form and language.

Thank you very much for your advice. We reworked the manuscript completely, regarding the language.
